# An Asymptomatic, Ectopic Mass as a Presentation of Adrenocortical Carcinoma Due to a Novel Germline *TP53* p.Phe338Leu Tetramerisation Domain Variant

**DOI:** 10.3390/children10111793

**Published:** 2023-11-07

**Authors:** Justyna Walenciak, Zuzanna Urbanska, Agata Pastorczak, Katarzyna Babol-Pokora, Kamila Wypyszczak, Ewa Bien, Aleksandra Gawlowska-Marciniak, Jozef Kobos, Wieslawa Grajkowska, Joanna Smyczynska, Wojciech Mlynarski, Szymon Janczar

**Affiliations:** 1Department of Pediatrics, Oncology and Haematology, Medical University of Lodz, 91-738 Lodz, Poland; justyna.walenciak@umed.lodz.pl (J.W.); zuzanna.urbanska@umed.lodz.pl (Z.U.); agata.pastorczak@umed.lodz.pl (A.P.); katarzyna.babol-pokora@umed.lodz.pl (K.B.-P.); kamila.wypyszczak@umed.lodz.pl (K.W.); wojciech.mlynarski@umed.lodz.pl (W.M.); 2Department of Pediatrics, Hematology and Oncology, Medical University of Gdansk, 80-210 Gdansk, Poland; ebien@gumed.edu.pl; 3Department of Pediatric Surgery and Oncology, Central University Hospital, Medical University of Lodz, 91-738 Lodz, Poland; aleksandra.gawlowska@umed.lodz.pl; 4Department of Normal and Clinical Anatomy, Chair of Anatomy and Histology, Medical University of Lodz, 92-213 Lodz, Poland; jozef.kobos@umed.lodz.pl; 5Department of Pathology, The Children’s Memorial Health Institute, 04-736 Warsaw, Poland; w.grajkowska@ipczd.pl; 6Department of Pediatrics, Endocrinology, Diabetology and Nephrology, Medical University of Lodz, 91-738 Lodz, Poland; joanna.smyczynska@umed.lodz.pl

**Keywords:** Li–Fraumeni syndrome, adrenocortical carcinoma, cancer predisposition, TP53

## Abstract

Adrenocortical carcinoma (ACC) is a rare cancer in childhood. ACC is frequently associated with germline *TP53* variants, with founder effects especially due to the p.Arg337His mutation. ACC leads to the secretion of adrenocortical hormones, resulting in endocrine syndromes, which is the usual trigger for establishing the diagnosis. We present a surprising ACC pathology in a non-secreting, ectopic retroperitoneal tumour in a 4-year-old boy, successfully controlled with chemotherapy and mitotane after microscopically incomplete tumour resection with spillage. Genomic analysis (gene panel sequencing and copy-number microarray) demonstrated a novel p.Phe338Leu tetramerisation domain (TD) *TP53* variant in the proband and his cancer-free mother and a monoallelic deletion encompassing the *TP53* locus in cancer tissue, consistent with cancer-predisposition syndrome. While the recurrent p.Arg337His variant translates into high ACC risk, residue 338 and, in general, TD domain variants drive heterogeneous clinical scenarios, despite generally being considered less disruptive than TP53 DNA-binding domain mutations.

## 1. Introduction

Adrenocortical carcinoma (ACC) is a rare cancer in childhood, constituting approximately 0.2% of all paediatric malignancies, with an incidence of 1 new case per 5 million children annually. ACC is characterised by two incidence peaks in childhood, the first under 3 years of age and the second in adolescence [1,2,3,4,5,6]. ACC is very frequently associated with germline pathogenic variants in the *TP53* tumour-suppressor gene consistent with Li–Fraumeni syndrome and thus might be much more prevalent in populations with founder effects, in particular due to the p.Arg337His ‘Brazilian’ variant [7,8,9,10]. ACC may also occur in the context of other cancer-predisposition syndromes, especially Lynch syndrome [7,11,12,13,14,15,16]. Usually, ACC leads to the excessive secretion of adrenocortical hormones resulting in Cushing’s syndrome, virilisation, or feminisation, which are the triggers for investigations and eventually diagnosis. Non-secreting tumours may occur in older children, which account for less than 10% of cases and are difficult to determine [1,16,17,18,19,20,21]. The data on ectopic ACC are very scarce and limited to case descriptions with the reported localisations including the bowel wall, ovaries, lungs, and spinal region [19,20,22,23]. The potential occurrence of a non-functional ACC in atypical/ectopic localisation largely precludes any pre-operative ACC suspicion unless there is a cancer-predisposition trait in the individual or the family. Furthermore, ectopic localisation compromises the potential assessment as per the Wieneke criteria, which is the best established adrenocortical tumour histopathological grading system, due to the difficult interpretation of at least two features of the scoring system (‘capsular invasion’ and ‘extension into periadrenal soft tissues and/or adjacent organs’) [1,24]. The prognosis in ACC is varied, depending on the tumour stage and other various prognostic features, some of which are not strongly established in children, and is consistently poor in metastatic disease, whereas there are too few data to establish the prognostic impact of non-functional or ectopic status [1,2,3,4,5,10,25]. TP53 germline-mutant status is suggested to be associated with favourable outcomes in ACC.

Surprisingly, in patients and families with the founder Brazilian Arg337His mutation, located in the tetramerisation domain and believed to be less disruptive than DNA-binding domain TP53 variants, ACC diagnosis age and the cancer spectrum in general are very heterogeneous. This is not well understood but might be dependent on other genomic variants such as, for example, the recently reported ADH7 variant [26].

## 2. Materials and Methods

### 2.1. Patient/Guardian Consent

The patient’s parents gave written consent to the conduct of the genetic studies and the publication of the results.

### 2.2. Next-Generation Sequencing (NGS)

Targeted NGS was performed using a custom-designed SureSelect QXT panel (Agilent Technologies Inc., Santa Clara, CA, USA), which included approximately 700 genes related to oncohaematological disorders. Sequencing libraries were prepared according to the manufacturer’s protocol. High-throughput sequencing was performed on NextSeq550 (Illumina, San Diego, CA, USA) using the Mid Output 300 bp paired-end run procedure (Illumina). The data analyses of the target regions were performed using Burrows–Wheeler Aligner Genome Alignment Software v2.1.2 and the GATK Variant Caller algorithms and mapped to the human genome reference sequence GRCh37/hg19 [2]. The results were then analysed using Variant Studio v. 3.0 (Illumina) and Integrative Genomics Viewer v.2.3. The filtering criteria included coverage with at least 20 reads and a minor allele frequency (MAF) below 0.01 in the GnomAD database. All filtered variants were investigated by several bioinformatics tools: SIFT, Mutation Taster, and PolyPhen-2. The pathogenicity of the revealed variations was estimated based on the ClinVar, OMIM, HGMD, and Varsome databases according to ACMG classification rules [27]. In addition, an internal database was used to filter out the recurrent variants.

### 2.3. DNA Sequencing

Sanger sequencing of TP53 (c.[1012T>C]) and ADH7 (rs971074) variants using DNA extracted from peripheral blood was performed using standard protocols.

### 2.4. Genomic Microarray Analysis

Copy number analysis of the tumour tissue was performed with the CytoScan XON array (Applied Biosystems, Thermo Fisher Scientific, Waltham, MA, USA), which contains 6.5 million markers for copy number analysis and approximately 300,000 SNPs. GeneChip Scanner 3000 (Thermo Fisher Scientific) was used. The assays were performed according to the current CytoScan XON assay user guide (version: 703456 Rev. 1). The final analysis was carried out using Chromosome Analysis Software (ChAS) v. 4.4 (Thermo Fisher Scientific). The aberrations are described in accordance with the ISCN 2020 recommendations.

## 3. Case Description

A 4-year-old boy, without previous relevant medical history, was admitted to hospital due to the incidental ultrasound finding of a small tumour of the left retroperitoneal area in proximity to the left kidney and spleen. Magnetic-resonance imaging (MRI) demonstrated a rounded mass anterior to, but separated from, the left renal upper pole of approximately 2 cm diameter, isointense to the muscles. It appeared separate from the adjacent spleen and bowel but difficult to separate from the pancreatic tail. There was no clear radiological impression of the tumour type and origin, and in particular, benign neuroblastic tumours were considered. The physical findings, including no cushingoid features and normal blood pressure, were without abnormalities. Also, the family history was non-contributory with virtually no cancer on the maternal side and cases of breast, colon, gastric, and prostate cancer in the father’s mother and grandparents at the age of 60 to 80. The tumour marker and endocrine work-up did not demonstrate increased neuron-specific enolase, alpha-fetoprotein, β-human chorionic gonadotrophin, urinary catecholamines, and their metabolites or any abnormalities in the profile of adrenal hormones and precursors (cortisol, testosterone, androstenedione, DHEA-S, 17-OH progesterone, estradiol, and aldosterone). The decision was to proceed with the primary resection of the tumour due to its unclear character and general impression of resectability. Figure 1A presents a 3D reconstruction of the computed tomography images of the tumour conducted in order to plan the resection. The tumour was resected laparoscopically. The surgical and histopathological reports were consistent with microscopically incomplete resection (‘tumour spillage’ due to rupture of the pseudomembrane and surgical margins positive for neoplastic cells). The somewhat surprising histopathological diagnosis, subsequently confirmed by two independent pathologists, was adrenocortical carcinoma (ACC). The ectopic location largely precluded assessment as per the Wieneke criteria; however, the notion of malignant character was mainly based on pseudocapsular invasion, extensive tumour necrosis, and atypical mitotic figures (Figure 1B demonstrates a microscopic image of a representative hematoxylin–eosin section). These surprising histopathological findings triggered more intensive post-operative work-up. The abdominal magnetic resonance (MR) and positron emission tomography (PET) imaging were both negative for any local residual disease, pathological lymph nodes, or the presence of a distant metastatic disease. The boy remained asymptomatic and with a normal adrenal endocrine profile. We initiated local routine NGS (next-generation sequencing)-based diagnostic custom-made panel sequencing of genes associated with cancer which yielded the *TP53* variant <NM_000546.6:c.[1012T>C]; NP_036584.1:p.[(Phe338Leu)]>, confirmed with direct sequencing, with potential pathogenicity predictions in 5 out of 7 metascore prediction tools (Appendix A). This variant has not been revealed so far in gnomAD genomes v. 2.1.1, nor in the internal laboratory database of 2204 alleles of individuals sequenced for various indications. According to the American College of Medical Genetics criteria, the germline variant is classified as “likely pathogenic” (ACMG criteria: PM1, PM2, and PP3). The same variant was revealed in the boy’s 42-year-old mother, who is cancer-free so far and with no suggestive family history (Sanger chromatogram shown in Figure 1C). Literature and database searches yielded three patient reports/submissions of other *TP53* variants in the same residue as discussed below (there are a few other entries in databases and publications but based on mutagenesis studies without related cancer cases and thus not included here).

We also carried out copy number analysis of the tumour tissue. This demonstrated a large 10.9 Mbp monoallelic deletion at *TP53* locus <17p13.3p12(5343_10944407)x1>, Figure 1D. The full molecular karyotype is provided in Appendix A. Furthermore, the proband and his mother were demonstrated to be both homozygotes for the common rs971074 ADH7 modifier gene allele.

Following diagnosis and staging, the patient was classified as stage III ACC and started adjuvant chemotherapy and mitotane as suggested by the diagnostic and therapeutic recommendations of the European Cooperative Study Group for Pediatric Rare Tumors (EXPeRT) [2]. Recently, he completed all six scheduled multidrug chemotherapy cycles, without major complications. The boy continues to take mitotane, with drug-level monitoring, and hydrocortisone substitution. So far, he is free of any signs of disease recurrence in MR and PET studies. The patient will enter a cancer-surveillance protocol, as proposed generally by the European Reference Network GENTURIS [28]. In our centre, this includes detailed physical examination including endocrine signs every 6 months, abdominal ultrasound every 3–4 months, and whole body and brain MRI annually. Initially, in our proband, the studies will be denser due to the recent completion of treatment and the risk of recurrence. In the mother, the scheme is similar but includes clinical breast examination twice a year and breast MRI annually. The proband’s mother was referred to a dedicated regional unit for adults with cancer-predisposition syndromes.

## 4. Discussion

In this report, we present a novel cancer-associated *TP53* (p.Phe338Leu) missense variant within the p53 tetramerisation domain (TD, residues 326–356) revealed in an asymptomatic pre-school boy following resection of a small retroperitoneal mass and a surprising pathological report of ACC. Mutations within the TD domain are not common but have already been described in Li–Fraumeni patients, including for example Arg337Cys, and in particular in paediatric-ACC/variant Li–Fraumeni (‘Brazilian mutation’ Arg337His). From a molecular point of view, while generally less disruptive than in other TP53 domains, mutations within the TD domain generally lead to compromised p53 tetramer stability and downstream consequences including perturbed DNA damage-induced signalling, MDM2 interaction, and DNA-binding activity and transcriptional activity [29]. It appears that in individuals with p53 TD domain mutations, the cancer penetrance is likely lower relative to DNA-binding domain TP53 mutations but varied and associated with heterogeneous cancer age and cancer spectrum. In particular, the well-documented Arg337His, frequently considered of incomplete penetrance, founder variant is expressed in some families as overt Li–Fraumeni syndrome, and in others, it is expressed as Li–Fraumeni-like syndrome or isolated ACC, the latter even without any family history of cancer [25,30]. This also likely seems to be the case with missense variants at the 338 residue. Apart from our proband with ACC and his so far cancer-free mother, the published reports include only two adult-onset cancer patients (p. Phe338Ser and p.Phe338Cys; no exact cancer data published) as well as an infant with p.Phe338Cys and choroid plexus carcinoma (CPC) [31]. We believe that our findings build up the catalogue of cancer-related *TP53* variants outside the DNA-binding domain, demonstrate the heterogeneity and varied penetrance, and motivate further variant reporting and clinical as well as functional research. From the practical and educational point of view of our report, the case supports considering ACC in the differential diagnosis of abdominal tumours. We conclude, based on our proband and the available literature, that there are no tools to predict cancer risk in patients with the p53 tetramerisation defect on an individual level, although high ACC risk is to be considered. We speculate that, in the future, modifier or marker genetic or epigenetic variants might be available, such as the proposed modifier ADH7 rs971074 common variant reported to modulate ACC cancer risk in Arg337His carriers [26] (this *ADH7* variant does not seem to be responsible for the discrepancy in the family we studied as both the proband and the mother are homozygous for the common rs971074 allele). Currently, we see no prospect of altering cancer surveillance programmes or adjusting genetic counselling for patients with TP53 TD domain variants as compared to patients with ‘classical’ mutations. Further studies are unlikely to change that perception due to the paucity of TD domain variants. 

## 5. Conclusions

We present a novel TP53 tetramerisation domain pathogenic variant so far revealed in a child with ACC with very atypical presentation. This p.Phe338Leu might be speculated to bear molecular and clinical similarities to founder p.Arg337His. Based on the minimal available literature as well as our report, cancer penetrance is difficult to predict.

## Figures and Tables

**Figure 1 children-10-01793-f001:**
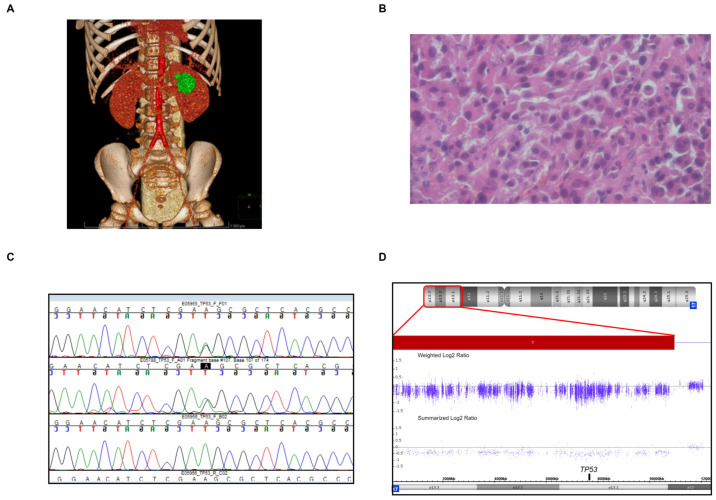
(**A**)—Three-dimensional reconstruction of the computed tomography images of the tumour; (**B**)—microscopic image of a representative hematoxylin–eosin section (400× magnification); (**C**)—Sanger sequencing chromatogram showing TP53 residue 337 variation in the proband and family members (upper row—proband, middle—mother, and bottom—father); (**D**)—CytoScan XON assay output from Chromosome Analysis Software (ChAS) v. 4.4 demonstrated a 10.9 Mbp monoallelic deletion encompassing the TP53 locus.

## Data Availability

The research data are available from the corresponding author upon reasonable request.

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
