# Peer review of "An Asymptomatic, Ectopic Mass as a Presentation of Adrenocortical Carcinoma Due to a Novel Germline *TP53* p.Phe338Leu Tetramerisation Domain Variant"

_children, 2023, doi:10.3390/children10111793_

Round 1

Reviewer 1 Report

Comments and Suggestions for Authors

In this study, the authors present a rare case of ACC-associated TP53 (p.Phe338Leu) missense variant within the p53 tetramerization domain (TD, residues 326–356) which was presented as an "accidentaloma" after abdomina US scan in a 4y boy and its diagnosis was confirmed only after the pathology analysis. It is an interesting case from its genetic perspective. However, regarding the clinical aspect of the case, the patient was handled in the usual way of assessing a new-diagnosed tumor. Interestingly, the diagnosis was established after the pathology analysis and all the genetic evaluations performed postoperatively.

Regarding the manuscript, some issues should be addressed.

The sentence "whereas there are no data to too few data to establish the 71 prognostic impact of non-functional or ectopic status" lines 71-72 should be rephrased: "no data to" should be omitted.

Line 82: "publica-tion" should change to "publication"

Nice presentation of the materials and methods followed for the genetic analysis of the case.

Lines 113-114: "Figure 1A presents 3D reconstruction 113 of the computed tomography images of the tumour." The authors should provide in precise way more details about the workup: what imaging tests performed, what were the findings of these tests, which was the differential diagnosis of the tumor preoperatively, why they decided to proceed to operation, if they thought that it may be a potential malignant tumor (ACC). Not just reporting a few words for the CT scan and an image. In addition, US images may be very helpful.

Finally, the authors should highlight the novelty of this case and what is its contribution to ACC management.

Comments on the Quality of English Language

Minor issues. 

Author Response

We would like to express our sincere gratitude to the reviewers for their time and effort devoted to assessing and improving our manuscript.

Below, we provide point-by-point information on how we dealt with the detected issues:

Reviewer 1

The sentence "whereas there are no data to too few data to establish the 71 prognostic impact of non-functional or ectopic status" lines 71-72 should be rephrased: "no data to" should be omitted.

The sentence was corrected as indicated.

Line 82: "publica-tion" should change to "publication"

The typo was corrected.

Lines 113-114: "Figure 1A presents 3D reconstruction 113 of the computed tomography images of the tumour." The authors should provide in precise way more details about the workup: what imaging tests performed, what were the findings of these tests, which was the differential diagnosis of the tumor preoperatively, why they decided to proceed to operation, if they thought that it may be a potential malignant tumor (ACC). Not just reporting a few words for the CT scan and an image. In addition, US images may be very helpful.

In the revised manuscript we clarify what diagnostic tests were performed preoperatively (ultrasound, MRI, CT). We added MRI description. Ultrasound images we can access are of too low quality for presentation. In the manuscript, we present 3D-CT reconstruction as it seemed most illustrative of all the modalities.

The undertaken imaging and laboratory tests were inconclusive, and there was no suspicion of ACC. The resection was undertaken to achieve the diagnosis and because of parents will versus for example observation. We did not consider biopsy because of general appearance of resectability.

Finally, the authors should highlight the novelty of this case and what is its contribution to ACC management.

The discussion is now altered to accommodate that request.

Reviewer 2 Report

Comments and Suggestions for Authors

This short case report describes the unusual presentation of a rare cancer in the context of a unique inherited TP53 mutation affecting its tetradimerisation domain. The case is very interesting and presentation of a new TP53 mutation adds to the catalogue of TP53 variants associated with inherited and malignant disease. The authors appear to have performed the appropriate laboratory tests in relation to this patient, and clinical management also seems to be appropriate. I recommend this manuscript for publication once the comments below have been addressed.

Questions and Comments

1. There are errors in the Figure 1 legend. Figures 1C and 1D are reversed. Please clearly label chromatograms from the proband and each family member (Sanger sequencing) and that the DNA source for each of the Sanger sequences is peripheral blood (if that was the case).

2. Do the tumour cells express TP53? (Have the authors performed TP53 immunohistochemistry?) If so, what was the pattern of TP53 immunostaining? This could be added to the manuscript text.

3. What surveillance will be provided for the proband’s family (the proband’s mother)? Brief details could be added to the manuscript.

Comments on the Quality of English Language

The manuscript is very well written. English language is very good, although there are many minor grammatical errors, mainly associated with omission of articles (‘the’ or ‘a’). I have listed some of these and other minor grammatical errors below, however the majority will need to be amended by an English language editor. 

Minor typographical errors

1. Line 50: ‘an incidence’

2. Line 55: ‘the p.Arg337His ‘Brazilian’ variant’

3. Line 75: ‘in the tetradimerization domain’

4. Line 78: ‘the recently reported’

5. Line 88: ‘in the process of’ is an incorrect phrase, but I am unable to determine what the intended meaning is.

6. Line 106: ‘The assays were performed’

7. ‘Line 116: ‘the maternal side’

8. Lines 118 and 131: work-up’

9. Line 124: ‘rupture’

10. Line 133: ‘residual disease’ (‘residue’ is incorrect)

11. Lines 206 and 207: ‘initiated’ (not initialized)

Author Response

We would like to express our sincere gratitude to the reviewers for their time and effort devoted to assessing and improving our manuscript.

Below, we provide point-by-point information on how we dealt with the detected issues:

Reviewer 2

  1. There are errors in the Figure 1 legend. Figures 1C and 1D are reversed. Please clearly label chromatograms from the proband and each family member (Sanger sequencing) and that the DNA source for each of the Sanger sequences is peripheral blood (if that was the case).

We corrected now these issues in the Legend and in the Methods Section.

  1. Do the tumour cells express TP53? (Have the authors performed TP53 immunohistochemistry?) If so, what was the pattern of TP53 immunostaining? This could be added to the manuscript text.

We did not perform TP53 IHC. While the initial ACC report was surprising to us, the subsequent two opinions were consistent and none of the, in total three, pathologist requested /performed p53 staining. We did not address the issue. This IHC is potentially possible to perform, if necessary, but not in the 10 day interval set for the revision. We would expect nuclear p53 accumulation as in the study by Ribeiro et al. of R337H patients (Proc Natl Acad Sci U S A. 2001 Jul 31; 98(16): 9330–9335.).

  1. What surveillance will be provided for the proband’s family (the proband’s mother)? Brief details could be added to the manuscript.

The surveillance in our country generally follows European guidelines as summarized in the work by Frebourg et al. (The European Reference Network GENTURIS. Guidelines for the Li–Fraumeni and heritable TP53-related cancer syndromes. Eur J of Human Genetics 2020 28:1379–1386). The reference was added.

The standard protocol in our centre include detailed physical examination including endocrine signs every 6 months, abdominal ultrasound every 3-4 months, whole body and brain MRI annually. Initially, in our proband, the studies will be denser due to recent accomplishment of the treatment.

In the mother the scheme is similar but include clinical breast examination twice a year and breast MRI annually. The proband’s mother was referred to dedicated unit for adults with cancer-predisposition syndromes, which is available in our region.

The information was added to the manuscript.

Spelling /grammar problems:

  1. Line 50: ‘an incidence’
  2. Line 55: ‘the p.Arg337His ‘Brazilian’ variant’
  3. Line 75: ‘in the tetradimerization domain’
  4. Line 78: ‘the recently reported’
  5. Line 88: ‘in the process of’ is an incorrect phrase, but I am unable to determine what the intended meaning is.
  6. Line 106: ‘The assays were performed’
  7. ‘Line 116: ‘the maternal side’
  8. Lines 118 and 131: work-up’
  9. Line 124: ‘rupture’
  10. Line 133: ‘residual disease’ (‘residue’ is incorrect)
  11. Lines 206 and 207: ‘initiated’ (not initialized)

Corrected as indicated